# Reasons and Solutions for the Decline in Model Performance after Editing

**Xiusheng Huang**[* 1,2,3]**, Jiaxiang Liu**[* 1,2]**, Yequan Wang**[† 3]
**Kang Liu**[† 1,2]
[1]The Key Laboratory of Cognition and Decision Intelligence for Complex Systems,
Institute of Automation, Chinese Academy of Sciences
[2]School of Artificial Intelligence, University of Chinese Academy of Sciences
[3]Beijing Academy of Artificial Intelligence, Beijing, China
huangxiusheng2020@ia.ac.cn, liujiaxiang21@mails.ucas.ac.cn,
tshwangyequan@gmail.com, kliu@nlpr.ia.ac.cn

## Abstract

Knowledge editing technology has received widespread attention for low-cost updates of incorrect or outdated knowledge in large-scale language models. However, recent research has found that edited models often exhibit varying degrees of performance degradation. The reasons behind this phenomenon and potential solutions have not yet been provided. In order to investigate the reasons for the performance decline of the edited model and optimize the editing method, this work explores the underlying reasons from both data and model perspectives. Specifically, 1) from a data perspective, to clarify the impact of data on the performance of editing models, this paper first constructs a **Multi-Q**uestion **D**ataset (**MQD**) to evaluate the impact of different types of editing data on model performance. The performance of the editing model is mainly affected by the diversity of editing targets and sequence length, as determined through experiments. 2) From a model perspective, this article explores the factors that affect the performance of editing models. The results indicate a strong correlation between the L1-norm of the editing model layer and the editing accuracy, and clarify that this is an important factor leading to the bottleneck of editing performance. Finally, in order to improve the performance of the editing model, this paper further proposes a **D**ump **for S**equence (**D4S**) method, which successfully overcomes the previous editing bottleneck by reducing the L1-norm of the editing layer, allowing users to perform multiple effective edits and minimizing model damage. Our code is available at https://github.com/nlpkeg/D4S.

## 1 Introduction

Large-scale language models (LLMs) have demonstrated exceptional performance in NLP tasks [Huang et al., 2021, 2022]. However, as knowledge continues to evolve, LLMs inevitably contain incorrect or outdated information. Due to their vast number of parameters, directly fine-tuning the model would require a substantial amount of computational resources [Gupta et al., 2023]. As a result, knowledge editing techniques have emerged as a low-cost and effective method for updating a model's knowledge [Yao et al., 2023]. These techniques involve modifying a small number of the model's parameters to update its internal knowledge [Ding et al., 2023]. However, growing evidence suggests that altering model parameters can have a negative impact on the model's performance. The

---

[*]Equal contribution.
[†]Corresponding authors.

38th Conference on Neural Information Processing Systems (NeurIPS 2024).

specific reasons behind this phenomenon remain unclear, and corresponding optimization methods are currently unavailable.

Previous research has investigated the performance degradation of edited models [Wang et al., 2023, Mazzia et al., 2023]. Specifically, Hase et al. [2024] found that edited models suffer from catastrophic forgetting, where they forget previously edited samples. Additionally, Gu et al. [2024] showed that edited models exhibit significant performance declines on downstream tasks, which severely hinders the practical applicability of knowledge editing techniques. However, these studies failed to identify the underlying causes of performance degradation in edited models and did not propose optimization methods to mitigate this issue [Zhang et al., 2024].

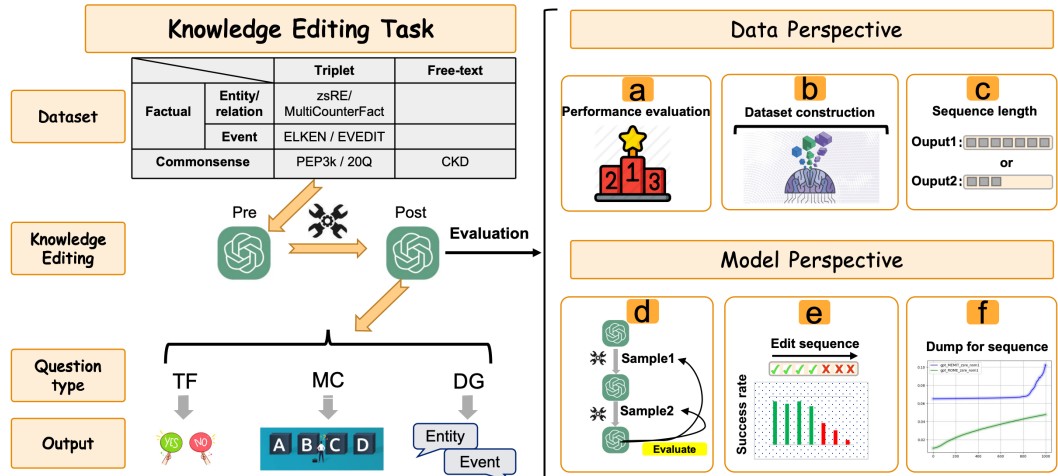

Fig. 1: This framework outlines the comprehensive approach to understanding the performance decline of edited models. On the left, traditional knowledge editing tasks are categorized into different types, each with distinct editing objectives: yes/no, a/b/c/d, and entity/event. On the right, our experiments are structured from both data and model perspectives. From the data perspective, we conduct three experiments: (a) a comprehensive performance evaluation of the model, (b) the construction of a **M**ulti-**Q**uestion **D**ataset (MQD), and (c) an assessment of the impact of editing different target outputs on model performance. From the model perspective, we design four experiments: (d) an evaluation of the edited model's forgetting ability, (e) an identification of the current knowledge editing method's bottleneck and an exploration of the correlation between editing probability values and parameter layer norms, and (f) a proposal of a sequence editing method, which effectively enhances the performance of the edited model.

This paper examines the factors that influence model performance and optimization methods from both data and model perspectives. As shown in Figure 1, from the data perspective, we evaluated the performance of edited models on multiple generalization tasks and found that the performance degradation of edited models is correlated with the editing objectives. We then constructed **M**ulti-**Q**uestion **D**ataset (MQD) with different question types, including multiple-choice, true/false, and direct generation, with corresponding editing objectives of yes/no, a/b/c/d, and entity/event, respectively. By calculating the perplexity (PPL) of different editing objectives, we discovered that the larger the PPL, the more severe the performance degradation of the edited model.

From the model's perspective, we investigate the reasons behind the decline in model performance from two angles: catastrophic forgetting and the bottleneck imposed by the number of edits. Our analysis reveals a strong correlation between the accuracy of edits and the L1-norm growth of the parameter layers after editing. To mitigate this issue, we propose a **D**ump **for S**equence (D4S) method that regulates the explosive growth of parameter layers during the editing process. This approach effectively enhances the performance of the edited model and overcomes the bottleneck associated with the number of edits.

To the best of our knowledge, we are the first to investigate the causes of performance degradation in edited models and concurrently propose an effective sequence editing method to enhance the performance of edited models. Our contributions can be summarized as follows:

- To investigate the impact of data on the performance of edited models, we performed evaluations across multiple tasks, revealing that the editing objective is the primary factor influencing model performance. By creating datasets with diverse question types, we established a strong correlation between the perplexity (PPL) of the editing objectives and the performance of the edited model.

- We find that the decline in edited model performance is correlated with the explosive growth of the L1-norm of parameter layers during the editing process, which is the primary cause of both catastrophic forgetting and the bottleneck imposed by the number of edits.

- To enhance the performance of the edited model, we propose a caching sequence edit method that leverages $\mathcal{O}(1)$ space complexity to retain past knowledge, regulates the explosive growth of parameter layer norms, and thereby effectively improves the performance of the edited model, ultimately overcoming the bottleneck imposed by the number of edits.

## 2 Related Work

### 2.1 Knowledge Editing Datasets

The existing knowledge editing dataset can be divided into triplet form and event form. In triplet format dataset, commonsense knowledge dataset includes PEP3k and 20Q [Porada et al., 2021, Gupta et al., 2023], factual knowledge includes ZsRE [Levy et al., 2017], CounterFact [Meng et al., 2022a], Fact Verification [Mitchell et al., 2022] , Calibration [Dong et al., 2022], MQuAKE [Zhong et al., 2023] and RaKE [Wei et al., 2023]. In event format dataset, datasets with only factual knowledge, including ELKEN [Peng et al., 2024], EVEDIT [Liu et al., 2024] and CKD[Huang et al., 2024].

### 2.2 Knowledge Editing Methods

The previous editing methods mainly focused on editing knowledge in the form of triples, with a small amount of knowledge in the form of editing events. The methods for editing triplet forms mainly include : (1)Locate-Then-Edit method [Dai et al., 2021, Meng et al., 2022a,b, Li et al., 2024], (2) Memory-based method [Mitchell et al., 2022, Madaan et al., 2022, Zhong et al., 2023, Zheng et al., 2023], (3) Hyper-network method [Mitchell et al., 2021, De Cao et al., 2021, Tan et al., 2023]. The method for editing event forms is Self-Edit [Liu et al., 2024].

### 2.3 Model Evaluation after Editing

The damage caused to the model by updating model parameters is unknown. [Hase et al., 2024] found that the edited model had catastrophic forgetting issues and performance degradation on downstream tasks. [Gu et al., 2024] evaluated the edited model on eight downstream tasks and used multiple editing methods. It was found that the edited model exhibited varying degrees of performance degradation. However, the above methods only found a decrease in performance of the edited model, without pointing out the reasons for the performance decline. At the same time, they did not propose effective editing methods to improve the performance of the edited model.

## 3 Data-Specific Factors Affecting Performance

In this section, we investigate the primary data factors contributing to the decline in model performance following editing.

### 3.1 The Overall Performance Evaluation

To assess the performance of the edited model, we curated a diverse range of editing and evaluation datasets for testing.

**The Dataset for Editing.** As illustrated in Figure 1, we selected common sense knowledge and factual knowledge datasets as editing corpora, featuring diverse data formats such as triplets and free text. In Figure 2, we utilized the zsRE[Levy et al., 2017], ELKEN[Peng et al., 2024], 20Q[Porada et al., 2021, Gupta et al., 2023], and CKD[Huang et al., 2024] datasets as editing corpora. Notably,

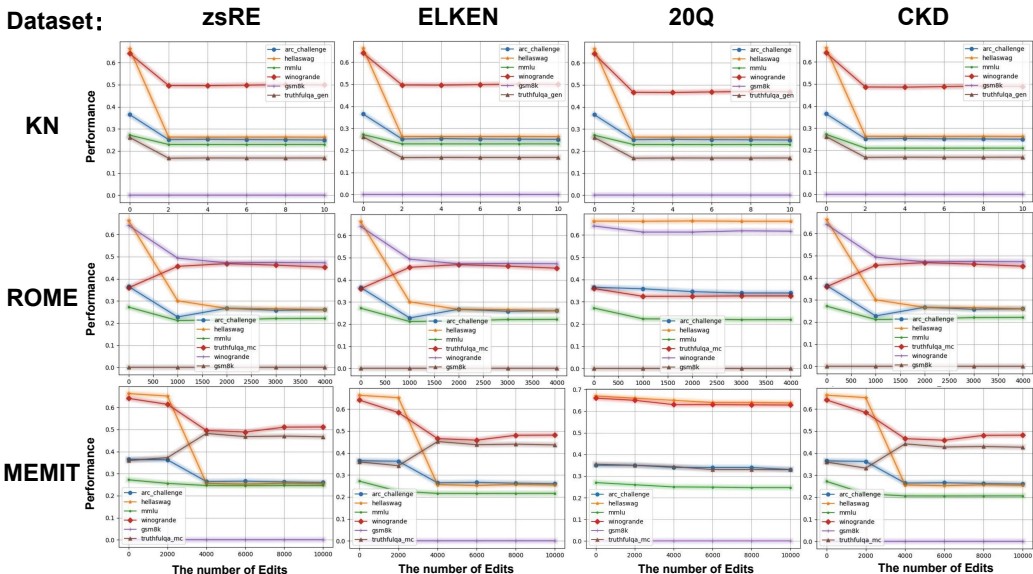

Fig. 2: Evaluation results of different editing methods on various types of datasets. The horizontal axis in the image represents the number of edited samples, and the vertical axis represents the performance of the edited model.

the commonsense dataset 20Q has editing objectives in the form of 0/1 labels, whereas the other three datasets have editing objectives of entity, entity, and event, respectively.

**The Dataset for Evaluation.** We employed six types of datasets in the Open LLM Leadership board evaluation, comprising AI2 Reasoning Challenge (25-shot) [Clark et al., 2018], HellaSwag (10-shot) [Zellers et al., 2019], the Common Sense Reasoning Dataset, MMLU (5-shot) [Hendrycks et al., 2020], a widely used benchmark for assessing multitask accuracy, which encompasses 57 tasks including basic mathematics, American history, computer science, law, and others. Additionally, we utilized TruthfulQA (0-shot) [Lin et al., 2021], a benchmark designed to test the model's propensity for dishonesty, as well as WinoGrande [Sakaguchi et al., 2021]: Common Sense Reasoning and GSM-8K [Cobbe et al., 2021]: Mathematical Reasoning.

**Result Analysis.** As illustrated in Figure 2, we observed that the KN [Dai et al., 2021] editing method substantially degrades the model's performance after editing a single sample. In contrast, when employing the ROME [Meng et al., 2022a] editing method, the model's performance did not experience a significant decline even after editing 1000 samples. Similarly, with the MEMIT [Meng et al., 2022b] method, the model's performance remained relatively stable when editing up to 2000 samples. However, beyond 2000 samples, a pronounced decline in performance occurred, and further increases in the number of edited samples did not lead to additional performance degradation.

It is noteworthy that the ROME [Meng et al., 2022a] and MEMIT [Meng et al., 2022b] methods exhibit less pronounced performance degradation when editing the 20Q [Porada et al., 2021, Gupta et al., 2023] dataset. In contrast, when editing other datasets, the model's performance displays varying degrees of decline. This can be attributed to the fact that the 20Q [Porada et al., 2021, Gupta et al., 2023] dataset's editing objective is in the form of binary labels (0/1), featuring a single element for the editing objective and a relatively small number of tokens.

### 3.2 Dataset Construction

To elucidate the impact of different editing objectives on the performance of the edited model, we created a **M**ulti-**Q**uestion **D**ataset (MQD) based on the ATOMIC commonsense database [Sap et al., 2019]. This dataset comprises three question types: true/false, multiple-choice, and direct generation. The corresponding editing objectives are yes/no, a/b/c/d, and entity/event, respectively. Each question type consists of 4000 samples.

| ATOMIC Data Source: < PersonX accepts PersonY appointment, as a result, PersonY shakes PersonX hand > | | |
| --- | --- | --- |
| **Category** | **Component Prompt** | **Target Answer** |
| DG | PersonX accepts PersonY appointment, resulting in PersonY | shakes PersonX hand |
| MQ | PersonX accepts PersonY appointment, resulting in PersonY ?
Below are four options: (a) to give him a treat; (b) to forgive him;
(c) to live happily ever after; (d) shakes PersonX hand. The correct option is | d |
| T/F | PersonX accepts PersonY appointment, resulting in PersonY
shakes PersonX hand . Is this sentence logical? Please answer yes or no. A: | yes |

Table 1: An example of converting source data from ATOMIC [Sap et al., 2019] database into directly generated(DG), multiple-choice questions(MQ), and true/false questions(T/F). The editing objectives are "shakes PersonX hand", "d" and "yes", respectively.

**Data Preparation.** The ATOMIC database [Sap et al., 2019], developed by the Allen Institute and later optimized in its subsequent version [Hwang et al., 2021], is a well-known commonsense repository. The data format in ATOMIC [Sap et al., 2019] is $< \text{Event}_1, \text{Relationship}, \text{Event}_2 >$, which contains unrecognized markers (e.g. ___, etc.) and invalid characters (e.g., &, etc.) that we manually filtered out. Furthermore, the relationship types in ATOMIC [Sap et al., 2019] are abbreviated and not easily comprehensible by humans. Although ATOMIC [Sap et al., 2019] provides corresponding annotations, they are insufficient to form a coherent statement when constructing the prompt. To address this, we also performed manual template rewriting to ensure a smoother overall prompt.

**MQD Dataset.** Different problem formats are associated with distinct editing objectives. The MQD dataset encompasses three formats: Directly Generated (DG), Multiple-choice Questions (MQ), and True/False questions (T/F). According to our statistical analysis, the Perplexity (PPL) values for the editing objectives of these three question types are 12.3, 43.3, and 297.4, respectively. The calculation formula for PPL is as follows:

$$\text{PPL}(X) = \exp\left\{ -\frac{1}{t} \sum_{i}^{t} \log p_\theta \left( x_i \mid x_{<i} \right) \right\} \tag{1}$$

Where $p_\theta \left( x_i \mid x_{<i} \right)$ is based on the sequence before i, and the log-likelihood of the i-th token. In the Table 1, for directly generating datasets, we directly concatenate $\text{Event}_1$, the rewritten relationship, and $\text{Event}_2$ to form a prompt. For multiple-choice questions, we designated the target answer as one of the options and randomly selected events from other natural categories in the dataset as the remaining options. For true/false questions, we asked LLMs to determine whether the newly formed prompt is logical. To mitigate bias, we established multiple positive and negative examples. Notably, the core information of the three question types remains consistent, with the primary difference lying in the question types and editing objectives.

### 3.3 The Influence of Editing Objectives

In this section, we investigate the effect of various editing objectives on model performance by editing the **M**ulti-**Q**uestion **D**ataset (MQD)3.2.

**The Dataset for Editing and Evaluation.** The MQD dataset comprises three question types: true/false, multiple-choice, and directly generated. The knowledge sources for these three question types are consistent, and we conducted relevant experiments directly using the MQD dataset. We selected ROME [Meng et al., 2022a] as the editing method. According to our statistical analysis, the average length of the input tokens for the three question types is 23.44, 35.03, and 13.38, respectively, while the average length of the editing objectives tokens is 1, 1, and 3.88, respectively. The true/false questions have two possible output types: yes or no. The multiple-choice questions have four editing objectives: a, b, c, and d. In contrast, the directly generated questions have more diverse editing objectives, including entities or events, with the number of tokens for events typically exceeding 1.

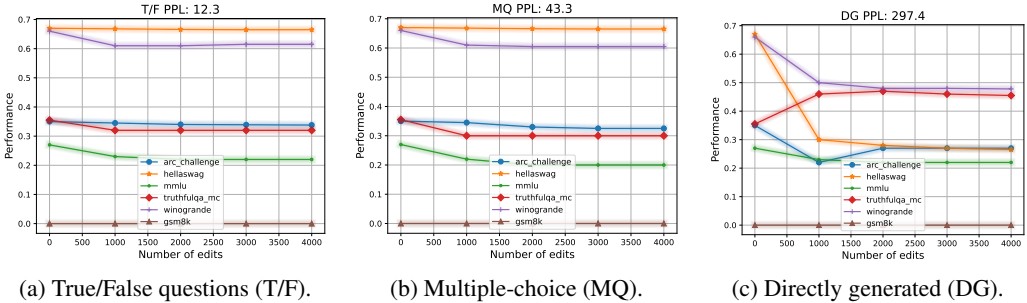

| (a) True/False questions (T/F). | (b) Multiple-choice (MQ). | (c) Directly generated (DG). |

Fig. 3: The performance of the model after editing data for different question types.

**Result Analysis.** As illustrated in Figure 3, all three question types - true/false, multiple-choice, and direct generation - compromise the model's performance, with direct generation causing the most significant damage. The experimental results suggest that a higher perplexity (PPL) of the editing objectives leads to more severe performance degradation. Meanwhile, we also calculated the L1-norm of the editing layer and found that the higher the perplexity of the editing target, the larger the L1-norm. Consequently, **from a data perspective, the decline in model performance after editing is attributed to the diversity of editing objectives and the length of tokens.**

## 4 Model-Specific Factors Affecting Performance

In this section, we investigate the model-centric factors contributing to performance degradation and propose optimization strategies to enhance the performance of the edited model.

### 4.1 Forgetting about Previously Edited Samples

As illustrated in Figure 7 in Appendix B, conventional editing methods typically assess a sample's quality immediately after editing. However, in real-world applications, it is often necessary to evaluate the editing success rate after processing a sequence of samples. The traditional evaluation approach overlooked the edited model's impact on previously edited samples. This section investigates the model's forgetting issue with respect to previously edited samples.

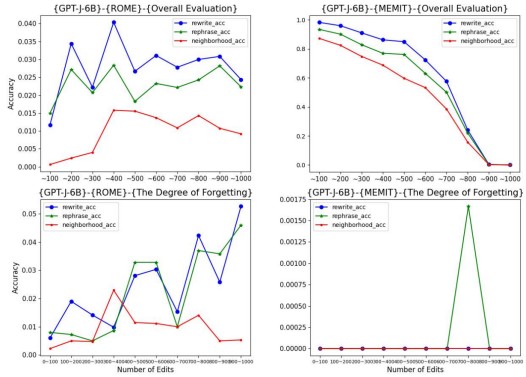

Fig. 4: Assessment of forgetting ability of models.

**The dataset and method for evaluation.** We employed the factual triplet dataset zeRE[Levy et al., 2017] for experimental purposes and utilized ROME [Meng et al., 2022a] and MEMIT [Meng et al., 2022b] as editing methods. To assess the varying degrees of forgetting in the model, we devised two experimental approaches. As illustrated in Figure 4, the "Overall Evaluation" involves evaluating all previous samples using the edited model each time it is completed. In contrast, "The Degree of Forgetting" involves obtaining a checkpoint after editing 1000 samples, followed by an evaluation of the model's forgetting degree every 100 previously edited samples, in the order of editing.

**Result analysis.** As shown in Figure 4, for the "Overall Evaluation", the ROME [Meng et al., 2022a] method has the highest accuracy of 0.04, indicating that the edited model has a significant forgetting problem on the 100 previously edited samples. The accuracy of the MEMIT [Meng et al., 2022b] method has decreased from the highest near 1.0 to 0.0, indicating that as the number of model edits increases, the forgetting problem of the model becomes increasingly severe. For the degree of forgetting, the ROME [Meng et al., 2022a] method and MEMIT [Meng et al., 2022b] method always have a low success rate. Especially the MEMIT [Meng et al., 2022b] method shows a success rate of 0 in most samples. We suspect that the MEMIT [Meng et al., 2022b] method after editing 1000

samples may not be able to successfully edit the samples. We also conducted relevant experiments in the next section.

## 4.2 The Bottleneck of Sequence Edit

Sequence editing refers to the repeated updating of knowledge within a model. However, the number of successful edits that existing knowledge editing methods can achieve is not infinite. We conducted experiments on two editing methods to examine the bottleneck in the number of edits for each method.

**The dataset and method for evaluation.** The editing models used in the experiments are all based on GPT-J (6B) [Wang and Komatsuzaki, 2021]. Consistent with the original work, the fixed MLP layers for the ROME [Meng et al., 2022a] and PMET [Li et al., 2024] methods are [5] and [3,4,5,6,7,8], respectively. We used the factual triplet dataset zeRE for experimental data and applied ROME and MEMIT as editing methods. All experiments were conducted with a batch size of 1, and a total of 1000 samples were edited. We recorded the probability value of each edited sample.

**Result analysis.** As shown in Figure 5.a, after editing 100 samples, the ROME method shows an overall decrease in probability values. While some probability values still update to larger values as the number of edits increases, the overall trend is downward. For the MEMIT method, after editing 850 samples, the probability values showed a significant decrease, approaching zero. This indicates that the MEMIT method exhibits a phenomenon of complete editing inefficiency after 850 edits. This experiment reflects the bottleneck phenomenon in editing methods, with ROME having an effective editing count of 100 and MEMIT having an effective editing count of 850.

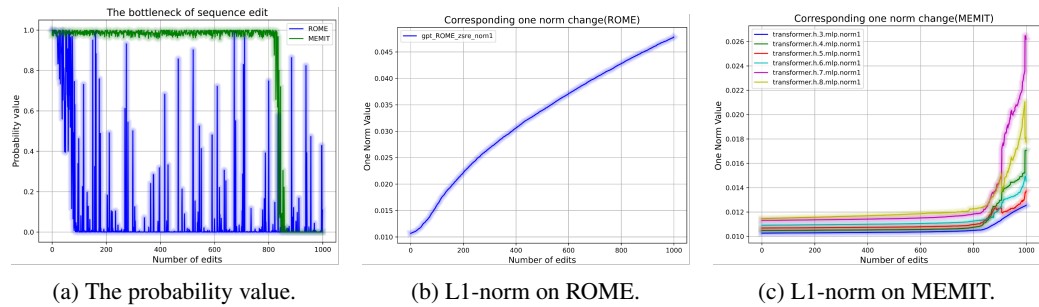

| (a) The probability value. | (b) L1-norm on ROME. | (c) L1-norm on MEMIT. |

Fig. 5: The bottleneck and L1-norm correspondence in sequence editing are illustrated in the figure. In subfigure (a), the horizontal axis represents the number of edited samples, and the vertical axis represents the probability value of the edited object. The blue line represents the ROME method, while the green line represents the MEMIT method. In subfigures (b) and (c), the horizontal axis represents the number of edits, and the vertical axis represents the L1-norm value of the editing layer.

We plotted the L1-norm variation of different methods on the editing layer during the editing process. As shown in Figure 5.b, when the number of edits reaches 100, the norm of the 5-th MLP layer edited by the ROME method significantly increases. Furthermore, in Figure 5.c, after 850 edits using the MEMIT method, the editing layers [3, 4, 5, 6, 7, 8] also exhibit an explosive increase in norm. The results indicate that **from a model perspective, the decline in model performance after editing is due to the explosive growth of norms in the editing layers during the editing process.**

## 4.3 Dump for Sequence Knowledge Editing

Sequence editing refers to the process of editing a single or multiple samples multiple times. With the continuous updating of world knowledge, constantly updating the knowledge within models has become an urgent need. The experimental results in Sections 3 and 4 show that the performance of the model significantly decreases after sequence editing. In this section, we propose a **D**ump **for Sequence (D4S) knowledge editing method that effectively improves the performance of the edited model.

### 4.3.1 Defects in Previous Sequence Editing

The knowledge editing method is to update factual associations stored in the parameters of transformers[Meng et al., 2022b]. Given a sequence of text $x = [x_1, \cdots, x_m]$, the transformer's hidden state $h^{l,j}$ at the layer $l$ and the token $j$ is calculated:

$$
\begin{aligned}
h^{l,j}[x] &= h^{l-1,j}[x] + att^{l,j}[x] + m^{l,j}[x] \\
att^{l,j}[x] &= attention^l(h^{l-1,1}[x], \cdots, h^{l-1,j}[x]) \\
m^{l,j}[x] &= W_{out}^l \sigma(W_{in}^l \gamma(att^{l,j}[x]))
\end{aligned}
\tag{2}
$$

where $\gamma$ indicate layer norm and $\sigma$ means a non-linear function. The knowledge editing method is to update the knowledge in the model by changing particular weight $W$. For MEMIT [Meng et al., 2022b] method, there are triples of fact to be edited $\xi = \{(s_i, r_i, o_i)\}_{i=1}^n$, where $s_i$ means the subject, $o_i$ is the object and $r_i$ means relation between them. And we have $o_i \neq LLM(s_i, r_i)$. For simplicity, we use $h_i^l[x]$ to represent the last token's hidden state ($h_i^{l,m}[x]$) of the $i^{th}$ prompt $x$. To make $o_i = LLM'(s_i, r_i)$, the target of $i^{th}$ edit $z_i = h_i^L + \delta_i$ is got by optimizing:

$$
\min_{\delta_i} \frac{1}{N} \sum_{k=1}^N -logP[o_i | pre_k \oplus p(s_i, r_i)]
\tag{3}
$$

where $h_i^L = h_i^L[p(s_i)]$ is the hidden state of the last edit layer $L$, $p(s_i, r_i)$ denotes the prompt consisting of subject and relation, and $pre_k$ indicates a random prefix to obtain generalizability. Subsequently, for each layer $l \in \mathcal{L}$ needs to be edited, we can take the following approach to update $W_{out}^l \in \mathcal{R}^{u \times v}$:

$$
k_i^l = \frac{1}{N} \sum_{k=1}^N \sigma(W_{in}^l \gamma(h_i^{l-1}[pre_k \oplus p(s_i)])), r_i^l = \frac{z_i - h_i^L}{L - l + 1}
\tag{4}
$$

$$
\Delta^l = (R^l K^{l^T})(Cov^l + K^l K^{l^T})^{-1}, W_{out}^l \leftarrow W_{out}^l + \Delta^l
\tag{5}
$$

with $R^l = [r_1^l, \cdots, r_n^l]$, $K^l = [k_1^l, \cdots, k_n^l]$, $Cov^l = K_0^l K_0^{l^T}$. And $K_0^l$ are the keys of knowledge irrelevant with $\xi$. A simple idea to optimize the knowledge of sequence editing as a whole is saving the editing history $R^l$ and $K^l$. For each new edit with $k_{n+1}^l$ and $r_{n+1}^l$, we can concate it with history:

$$
R^{l'} = R^l \oplus r_{n+1}^l = [r_1^l, \cdots, r_n^l, r_{n+1}^l]
\tag{6}
$$

$$
K^{l'} = K^l \oplus k_{n+1}^l = [k_1^l, \cdots, k_n^l, k_{n+1}^l]
\tag{7}
$$

In this way, we can optimize all the knowledge of sequence editing as a whole to mitigate the damage to LLM. However, the space complexity of such a dump method is $\mathcal{O}(n)$, which is unacceptable to us.

### 4.3.2 The D4S Method

The D4S method is designed to save the editing history in $\mathcal{O}(1)$ space and apply batch editing methods in sequence editing situations. We note that for $\Delta^l = (R^l K^{l^T})(Cov^l + K^l K^{l^T})^{-1}$, the parts related to the editing history can be written as:

$$
R^l K^{l^T} = [r_1^l, \cdots, r_n^l][k_1^l, \cdots, k_n^l]^T = \sum_{i=1}^n r_i^l k_i^{l^T}
\tag{8}
$$

$$
K^l K^{l^T} = [k_1^l, \cdots, k_n^l][k_1^l, \cdots, k_n^l]^T = \sum_{i=1}^n k_i^l k_i^{l^T}
\tag{9}
$$

where we have $R^l K^{lT} \in \mathcal{R}^{u \times v}$ and $K^l K^{lT} \in \mathcal{R}^{v \times v}$. So we just need to save the two matrices above. For each new edit with $k_{n+1}^l$ and $r_{n+1}^l$, we can integrate it into edit history with a smiple addition operation:

$$R^l K^{lT'} = R^l K^{lT} + r_{n+1}^l {k_{n+1}^l}^T \tag{10}$$

$$K^l K^{lT'} = K^l K^{lT} + k_{n+1}^l {k_{n+1}^l}^T \tag{11}$$

This approach requires just $\mathcal{O}(1)$ storage space and allows us to convert sequence editing methods into batch editing methods, thus reducing the damage to the edited model during sequence editing. Additionally, this ability to consolidate each individual edit instance into a single batch makes the locate and editing method distinct from fine-tuning.

### 4.3.3 Theoretical Proof of Mitigating Norm Growth

To demonstrate that our D4S method can effectively alleviate norm growth in the editing layer, we can consider the update of parameters edited by the previous method MEMIT [Meng et al., 2022b] after $n$ edits:

$$\Delta W_{MEMIT} = \sum_{i=1}^{n} (r_i k_i^T)(K_0 K_0^T + k_i k_i^T)^{-1} \tag{12}$$

Regarding the D4S method, we have:

$$\Delta W_{D4S} = (\sum_{i=1}^{n} r_i k_i^T)(K_0 K_0^T + \sum_{i=1}^{n} k_i k_i^T)^{-1} = \sum_{i=1}^{n} (r_i k_i^T)(K_0 K_0^T + \sum_{i=1}^{n} k_i k_i^T)^{-1} \tag{13}$$

Due to $K_0 K_0^T + k_i k_i^T$ and $K_0 K_0^T + \sum_{i=1}^{n} k_i k_i^T$ being positive definite, intuitively, the inverse of $K_0 K_0^T + \sum_{i=1}^{n} k_i k_i^T$ is expected to have smaller numerical values compared to $K_0 K_0^T + k_i k_i^T$. Therefore, the norm of $\Delta W_{D4S}$ is smaller than that of $\Delta W_{MEMIT}$. The experimental results in Figures 6 also demonstrate the effectiveness of the D4S method in mitigating L1-norm growth.

## 5 Experiments

The experiments are designed to answer two research questions: a) How does D4S perform in sequence editing compared to other methods? b) How much damage does D4S do to the model?

### 5.1 Performance Comparison of Sequence Editing

We counted the results of different methods after 1,000 sequence edits on GPT-J (6B) [Wang and Komatsuzaki, 2021] and Llama2 (7B) [Touvron et al., 2023]. The Appendix C provides Metric indicators. As show in Table 2, compared to previous editing methods, our method D4S has achieved significant performance improvement. Specifically, when Edits=500 and Edits=1000, the average performance of D4S achieved State-Of-The-Art(SOTA) results, indicating that D4S still has superior editing ability even after editing multiple samples.

We conduct additional experiments on Counterfact[Meng et al., 2022b] and Mquake[Zhong et al., 2023] dataset in Appendix D. In addition to this, we also wanted to explore the model's forgetting of previous edits, so we chose every 100 edits as a checkpoint to investigate this in Appendix E.

| Model | Method | Edits = 100 | | | | Edits = 500 | | | | Edits = 1000 | | | |
|---|---|---|---|---|---|---|---|---|---|---|---|---|---|
| | | *Eff.* | *Par.* | *Spe.* | *Avg.* | *Eff.* | *Par.* | *Spe.* | *Avg.* | *Eff.* | *Par.* | *Spe.* | *Avg.* |
| Llama | FT | 30.47 | 25.78 | 12.19 | 22.81 | 19.43 | 18.06 | 6.56 | 14.68 | 16.13 | 13.89 | 5.96 | 12.00 |
| | ROME | 97.68 | **92.39** | 90.22 | **93.43** | 48.68 | 47.72 | 32.56 | 42.99 | 21.84 | 20.74 | 4.01 | 15.53 |
| | MEMIT | 92.99 | 83.22 | 94.02 | 90.08 | 2.90 | 2.90 | 2.51 | 2.77 | 2.85 | 2.85 | 2.74 | 2.82 |
| | PMET | 0.24 | 0.35 | 0.46 | 0.35 | 0.00 | 0.00 | 0.00 | 0.00 | 0.00 | 0.00 | 0.00 | 0.00 |
| | GRACE | **99.33** | 0.53 | **99.88** | 66.58 | **98.87** | 0.59 | **99.72** | 66.39 | **98.94** | 0.35 | **99.77** | 66.35 |
| | **D4S**(ours) | 95.53 | 86.45 | 93.44 | 91.81 | 91.19 | **81.48** | 83.34 | **85.34** | 87.36 | **79.30** | 78.99 | **81.88** |
| GPT | FT | 15.28 | 7.40 | 0.42 | 7.70 | 12.95 | 7.84 | 0.23 | 7.01 | 7.03 | 4.15 | 0.11 | 3.77 |
| | ROME | 1.17 | 1.50 | 0.06 | 0.91 | 2.67 | 1.83 | 1.55 | 2.02 | 2.44 | 2.23 | 0.92 | 1.86 |
| | MEMIT | 98.25 | 93.41 | 87.24 | 92.97 | 84.94 | 76.15 | 59.72 | 73.60 | 0.00 | 0.02 | 0.00 | 0.01 |
| | PMET | 0.33 | 0.33 | 0.00 | 0.22 | 0.00 | 0.00 | 0.00 | 0.00 | 0.00 | 0.00 | 0.00 | 0.00 |
| | GRACE | **100.00** | 0.40 | **100.00** | 66.8 | **100.00** | 0.15 | **100.00** | 66.72 | **100.00** | 0.07 | **100.00** | 66.69 |
| | **D4S**(ours) | 97.72 | **97.01** | 85.30 | **93.34** | 97.42 | **93.12** | 74.65 | **88.40** | 95.98 | **91.17** | 70.17 | **88.77** |

Table 2: Sequencial edits on GPT and Llama with *ZsRE* dataset. The best results are in **bold** and underline means the suboptimal.

## 5.2 Performance of Edited Model

As shown in Figure 6, we explore the impact of D4S on the model by examining the changes in average weight norms and performance on downstream tasks of GPT. The downstream task performance changes for Lllama are in the Appendix F. The experimental results indicate that the norm of the D4S method did not increase after 10000 edits, and the performance of downstream tasks only decreased slightly.

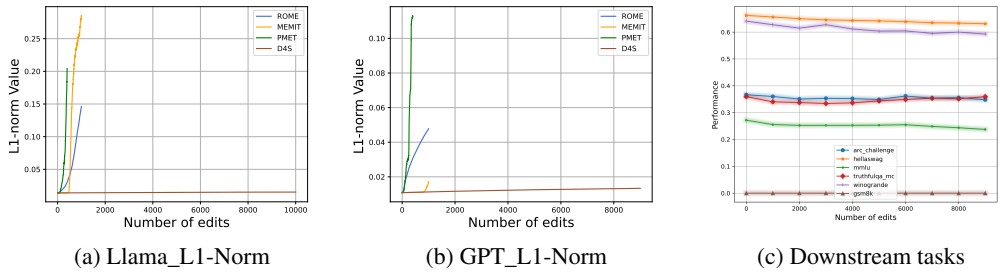

| (a) Llama_L1-Norm | (b) GPT_L1-Norm | (c) Downstream tasks |
|---|---|---|

Fig. 6: Norms of weight and performance of the edited model.

## 6 Conclusion

This paper explores the reasons behind the decline in model performance from both data and model perspectives. From the data perspective, the decline is attributed to the diversity of editing objectives and the length of tokens. From the model perspective, the decline is due to the explosive growth of norms in the editing layers during the editing process. To enhance the performance of edited models, we propose a **D**ump **for S**equence (D4S) method, which effectively improves the performance of edited models and overcomes previous editing bottleneck issues. This method allows for multiple effective edits with minimal impact on model performance.

## Acknowledgments

This work was supported by Beijing Natural Science Foundation (L243006) and the National Science Foundation of China (No. 62106249). This work was also sponsored by CCF-BaiChuan-Ebtech Foundation Model Fund.

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

# A    Appendix /Limitations

We note a few limitations of the experiments conducted in this paper:

(1) This paper only used the GPT-J (6B) and Llama2 (7B) models. Due to limitations in computational resources, experiments on larger scale models were not conducted.

(2) For the D4S method, our training consists of only 2000 steps due to computational resource limitations. By editing more samples, we can better identify the performance bottleneck of the D4S method.

(3) Due to computational resource limitations, the MEMIT and PMET methods did not use a batch size greater than 1 in our experiments. To better assess batch editing, more experiments with larger batch sizes should be attempted.

# B    Appendix /Evaluation Method

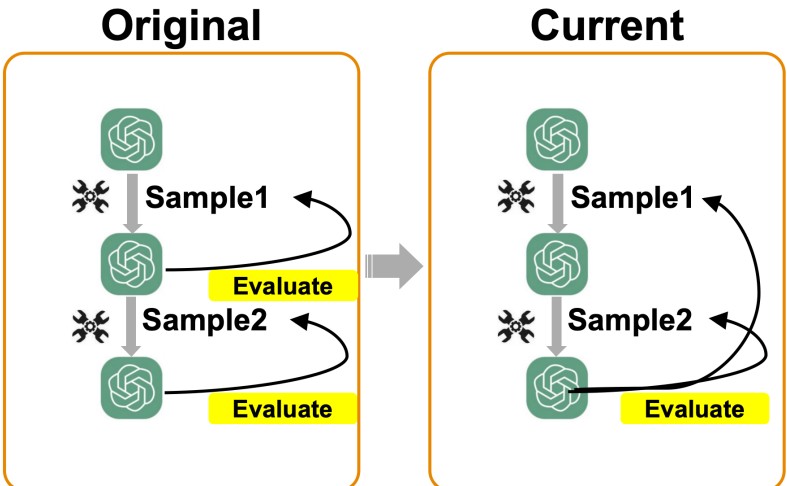

Fig. 7: The original and current evaluation methods.

# C    Appendix /Metric Indicators

Same as MEMIT, we took the following three indicators to evaluate the impact on model performance:

$$Eff. = \mathbb{E}_i[o_i = \underset{o_i^*}{argmax}\, P[o_i^*|p(s_i, r_i)]] \tag{14}$$

$$Par. = \mathbb{E}_i[\mathbb{E}_{p\in par(s_i,r_i)}[o_i = \underset{o_i^*}{argmax}\, P[o_i^*|p]] \tag{15}$$

$$Spe. = \mathbb{E}_i[\mathbb{E}_{(s_i^e,r_i^e,o_i^e)\in nei(s_i,r_i,0_i)}[o_i^e = \underset{o_i^*}{argmax}\, P[o_i^*|p(s_i^e, r_i^e)]] \tag{16}$$

where $par(s_i, r_i)$ is the set of $p(s_i, r_i)$'s paraphrases, $nei(s_i, r_i, o_i)$ is the set of $(s_i, r_i, o_i)$'s neighborhoods.

# D    Appendix /Additional Experiments

We also conduct additional experiments on Counterfact[Meng et al., 2022b] and Mquake[Zhong et al., 2023] dataset. Here are the results:

| Model | Method | Counterfact | | | | Mquake | | | |
|---|---|---|---|---|---|---|---|---|---|
| | | $Eff.$ | $Par.$ | $Spe.$ | $Avg.$ | $Eff.$ | $Par.$ | $Spe.$ | $Avg.$ |
| GPT | FT | 32.80 | 9.00 | 1.00 | 14.27 | 17.00 | 6.22 | 0.00 | 7.74 |
| | ROME | 0.60 | 0.70 | 0.60 | 0.63 | 0.00 | 0.00 | 0.00 | 0.00 |
| | MEMIT | 86.20 | **59.50** | 31.30 | 59.00 | 0.00 | 0.00 | 0.00 | 0.00 |
| | GRACE | **100.00** | 0.50 | **100.00** | 66.83 | - | - | - | - |
| | **D4S**(ours) | 99.10 | 47.00 | 63.70 | **69.93** | **97.40** | **75.54** | **21.84** | **64.93** |
| Llama | FT | 8.46 | 4.07 | 2.03 | 4.85 | 41.43 | 44.93 | **28.24** | 38.20 |
| | ROME | 27.83 | 16.03 | 5.66 | 16.50 | 76.85 | **78.41** | 3.67 | 52.97 |
| | MEMIT | 0.00 | 0.00 | 6.72 | 2.24 | 0.00 | 0.00 | 0.00 | 0.00 |
| | GRACE | **99.9** | 0.25 | **99.97** | 66.71 | - | - | - | - |
| | **D4S**(ours) | 96.68 | **46.66** | 72.45 | **71.93** | **85.30** | 72.68 | 28.16 | **62.05** |

Table 3: Sequential edits on GPT and Llama with *Counterfact* and *Mquake* dataset. The best results are in **bold** and underline means the suboptimal. Since GRACE is not suitable for the data structure of *Mquake*, we did not include it in the comparison.

# E Appendix /Evaluate Forgetting Ability

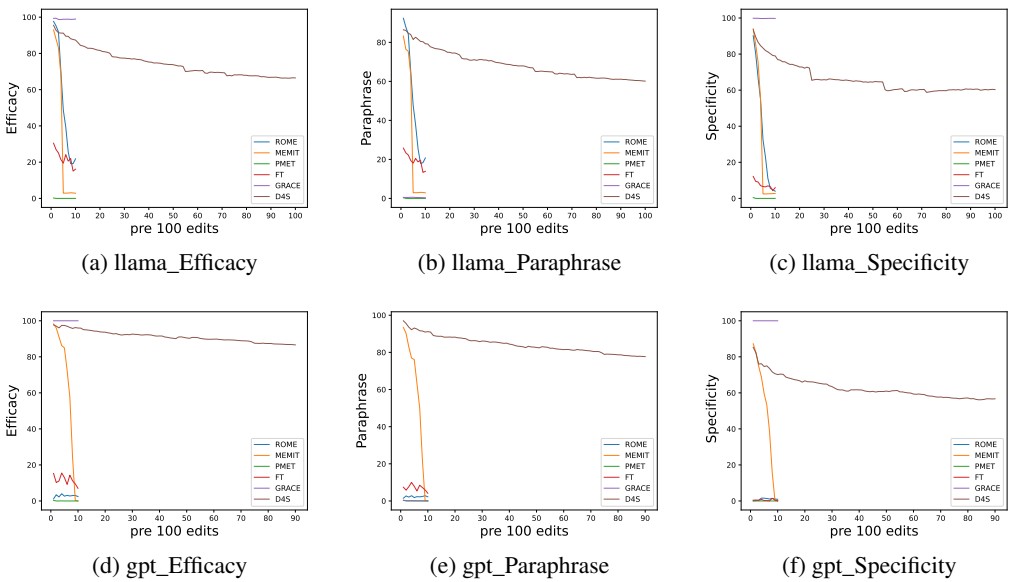

(a) llama_Efficacy     (b) llama_Paraphrase     (c) llama_Specificity

(d) gpt_Efficacy     (e) gpt_Paraphrase     (f) gpt_Specificity

Fig. 8: Evaluation metrics change with the number of edits.

# F Appendix /Downstream Performance of Llama

| Edit Num | arc_challenge | hellaswag | mmlu | truthfulqa_mc | winogrande |
|---|---|---|---|---|---|
| 0 | 43.34 | 57.14 | 41.35 | 38.97 | 69.14 |
| 10,000 | 42.41 ($\downarrow$0.93) | 52.99 ($\downarrow$4.15) | 39.76 ($\downarrow$1.59) | 38.56 ($\downarrow$0.41) | 68.75 ($\downarrow$0.39) |

Table 4: Downstream performance of llama after 10,000 edits.

