# OpenReview forum: "Reasons and Solutions for the Decline in Model Performance after Editing"
_NeurIPS.cc/2024/Conference — NeurIPS 2024 poster_

### Official Review · Reviewer_Kq3N · 2024-07-07

**Soundness:** 3
**Presentation:** 2
**Contribution:** 3
**Rating:** 6
**Confidence:** 3

**Summary:**

This paper addresses the challenges associated with the decline in performance of LLMs after undergoing knowledge editing. The study identifies the primary factors contributing to performance degradation from both data and model perspectives. By constructing a Multi-Question Dataset (MQD) and analyzing the impact of editing objectives, token length, and diversity, the paper finds that perplexity associated with editing objectives significantly affects model performance. From the model perspective, a strong correlation was observed between the L1 norm of parameter layers and editing accuracy. The paper proposes a novel method called Dump for sequence (D4C), which effectively manages the parameter growth and improves model performance post-editing.

**Strengths:**

- Innovative Methodological Approach: The study introduces a new method, D4C, which addresses the explosive growth in parameter norms and optimizes model performance post-editing. This approach is both innovative and practical for managing edited models.
- Comprehensive Data Analysis: The construction of the Multi-Question Dataset and detailed analysis of how different types of data affect model performance provide valuable insights into the mechanics of model editing.
- Clear Identification of Problems and Solutions: The paper clearly identifies specific problems associated with knowledge editing in LLMs, such as catastrophic forgetting and performance bottlenecks, and provides targeted solutions to these issues.
- Empirical Validation: The experiments conducted in this paper offer empirical evidence supporting the proposed methods, enhancing the credibility and applicability of the findings.

**Weaknesses:**

- Generalizability of Findings: The study focuses on specific scenarios and datasets, which may limit the generalizability of the findings across different types of LLMs or editing tasks.
- Potential Overfitting to Edited Scenarios: There is a risk that the model may become overly optimized for the edited scenarios, potentially affecting its performance on unedited or unrelated tasks.
- Complexity of Implementation: The proposed D4C method, while effective, may be complex to implement and integrate into existing systems due to its sophisticated handling of parameter layers.
- Unsuitable Citation Format: The citations in this paper are in the format of “XXX et al. [YEAR]”, which are not suitable enough, and had better change into the format of [1], [2], [3], ……

**Questions:**

- Adaptability of D4C Method: How adaptable is the D4C method to different types of LLMs and knowledge editing tasks beyond those tested in your experiments?
- Impact on Unedited Model Performance: How does the D4C method affect the performance of the model on tasks that have not been edited? Is there any evidence of performance trade-offs?
- Handling of Diverse Editing Objectives: Could you elaborate on how the D4C method manages the complexity and diversity of editing objectives without compromising the model’s overall integrity and coherence?

**Missing References**
- Editing Large Language Models: Problems, Methods, and Opportunities (EMNLP 2023)
- Knowledge Editing for Large Language Models: A Survey (2023)
- A Survey on Knowledge Editing of Neural Networks (2023)
- A Comprehensive Study of Knowledge Editing for Large Language Models (2024)

**Limitations:**

- Dependency on Specific Data Characteristics: The effectiveness of the proposed solutions may depend heavily on the characteristics of the data used for training and testing, which might not be consistent across different domains or applications.
- Evaluation Metrics: While the paper introduces new evaluation methods, the reliance on perplexity (PPL) and L1 norm metrics might not completely capture all aspects of model performance and health, especially in nuanced or context-dependent scenarios.
- Limited Experimentation: The experiments (Section 5) in this paper are too limited to demonstrate the conclusion.
- Scope of Editing Objectives: The study might not fully capture the impact of highly diverse or complex editing objectives that could be encountered in real-world scenarios.

---

> ### Author Rebuttal · Authors · 2024-08-05
>
> Thank you for your constructive feedbacks on the paper! We have added detailed explanations for the important questions asked in the review.
>
> $\textbf{Q1: }$ How adaptable is the D4C method to different types of LLMs and knowledge editing tasks beyond those tested in your experiments?
>
> $\textbf{W1:}$ We conducted additional experiments and expanded the datasets to include the Mquake [1] and Counterfact [2] datasets. The experimental results on GPT-J are as follows:
>
> |GPT-J-Mquake |Eff.|Par.|Mul.|Avg.|
> |----  |----  |----  |----  |----  |
> |FT|`17.00`|`6.22`|`0.00`|`7.74`|
> |ROME|0.00|0.00|`0.00`|0.00|
> |MEMIT|0.00|0.00|`0.00`|0.00|
> |D4C|**97.40**|**75.54**|**21.84**|**64.93**|
>
> |GPT-J-Counterfact |Eff.|Par.|Spe.|Avg.|
> |----  |----  |----  |----  |----  |
> |FT|32.80|9.00|1.00|14.27|
> |ROME|0.60|0.70|0.60|0.63|
> |MEMIT|86.20|**59.50**|31.30|59.00|
> |GRACE|**100.00**|0.50|**100.00**|`66.83`|
> |D4C|`99.10`|`47.00`|`63.70`|**69.93**|
>
> At the same time, we extended the model to Llama2, and the results are shown below:
>
> |Llama2-Mquake |Eff.|Par.|Mul.|Avg.|
> |----  |----  |----  |----  |----  |
> |FT|41.43|44.93|**28.24**|38.20|
> |ROME|`76.85`|**78.41**|3.67|`52.97`|
> |MEMIT|0.00|0.00|0.00|0.00|
> |D4C|**85.30**|`72.68`|`28.16`|**62.05**|
>
>
> |Llama2-Counterfact |Eff.|Par.|Spe.|Avg.|
> |----  |----  |----  |----  |----  |
> |FT|8.46|4.07|2.03|4.85|
> |ROME|27.83|`16.03`|5.66|16.50|
> |MEMIT|0.00|0.00|6.72|2.24|
> |GRACE|**99.9**|0.25|**99.97**|`66.71`|
> |D4C|`96.68`|**46.66**|`72.45`|**71.93**|
>
> PS: **Bold** indicates the best results, while suboptimal is `highlighted`.
>
> The results show that our method has superior performance on different types of datasets and models. Furthermore, We have expanded our experimental editing 10,000 times, and our performance far exceeds other methods. **You can find the details in the supplementary PDF  of Global Response.**
>
> $\textbf{Q2: }$  How does the D4C method affect the performance of the model on tasks that have not been edited? Is there any evidence of performance trade-offs?
>
>
> $\textbf{W2: }$  As highlighted in Section 4.2, we observed a strong correlation between the rise in the L1-norm of the parameter layer and the performance of the editing task. Simultaneously, we noted a significant decrease in the performance of the edited model on unedited tasks. We posit that the increase in the L1-norm results in decreased performance on tasks that have not undergone editing. Our approach effectively mitigates the norm growth of the edited model by consolidating all samples edited from the sequence (e.g., 6.a and 6.b). A succinct theoretical proof is presented in Reviewer MFeF's 'Q6'. Figure 6.c demonstrates that the D4C method ensures that the edited model maintains superior performance on unedited tasks.
>
> Furthermore, we extended the number of edits to 10,000, and the edited model still maintained good performance.
>
> |Num of Edits|arc_challenge|hellaswag|mmlu|truthfulqa_mc|winogrande |
> |---- |----  |----  |----  |----  |----  |
> |0|43.34|57.14|41.35|38.97|69.14|
> |10,000|42.41 ($\downarrow 0.93$)|52.99 ($\downarrow 4.15$)|39.76 ($\downarrow 1.59$)|38.56 ($\downarrow 0.41$)|68.75 ($\downarrow 0.39$)|
>
> The existing knowledge editing methods have poor editing performance (such as Table 2), or have a storage complexity of $O(n)$ ( e.g. GRACE[3]), and cannot provide experimental results edited 10,000 times.
>
>
> $\textbf{Q3: }$ Could you elaborate on how the D4C method manages the complexity and diversity of editing objectives without compromising the model’s overall integrity and coherence?
>
> $\textbf{W3: }$  The complexity and diversity of editing objectives can affect the performance of the edited model, which is the conclusion we have drawn from a data perspective. Our further experiments indicate a correlation between editing objectives and parameter layer norm growth. For example, if the editing objective for a true/false question is "yes/no", the norm growth caused by this dataset is slow, while the directly generated editing objective is "entity/event", the norm growth caused by this dataset is fast. This also confirms our conclusion from a model perspective that the reason for the performance degradation of the edited model is related to norm growth. We will present the specific experimental results in the revised version.
>
> Our D4C method can effectively reduce the norm growth of the edited model (such as Fig 6.a and 6.b), even when the number of edits reaches 10,000 (e.g. W2). Therefore, D4C can manage the complexity and diversity of editing objectives without compromising the model’s overall integrity and coherence.
>
> $\textbf{Q4: }$ The proposed D4C method, while effective, may be complex to implement and integrate into existing systems due to its sophisticated handling of parameter layers.
>
> $\textbf{W4: }$ Thank you for recognizing the effectiveness of our method. To be honest, our system is not complex, with core code ranging from 200 to 300 lines. We have included our code in the Supplementary Material when submitting the paper. You can download the file and follow the instructions in the README.md to set up the code.
>
> $\textbf{Ref}$
>
> [1] Mquake: Assessing knowledge editing in language models via multi-hop questions. EMNLP 2023.
>
> [2] Mass-editing memory in a transformer. ICLR 2023.
>
> [3] Aging with GRACE: Lifelong Model Editing with Discrete Key-Value Adaptors. Neurips 2023.

---

> > ### Comment · Reviewer_Kq3N · 2024-08-13
> >
> > Dear Authors,
> >
> > Thank you very much for the clarification! Most of them have addressed my concerns.
> >
> > I appreciate the response, especially the newly conducted experiments. Including those in the revised draft will strengthen the paper.
> >
> > Overall, I acknowledge the interesting idea of this work, and decide to increase my original scores 5 to 6.
> >
> > Best Regards,
> >
> > Reviewer Kq3N

---

> > > ### Author Response · Authors · 2024-08-14
> > >
> > > Thank you for your reply. We promise to add additional experiments in the revised version.

---

### Official Review · Reviewer_MFeE · 2024-07-09

**Soundness:** 3
**Presentation:** 4
**Contribution:** 3
**Rating:** 8
**Confidence:** 4

**Summary:**

Recent research has shown varying degrees of decline in model performance following small changes made by certain model editing methods. This paper is the first to comprehensively analyze the reasons behind such performance declines. Through extensive experiments, it identifies two main factors: data and model. For data-specific factors, the paper finds that perplexity and token length significantly influence performance. For model-specific factors, the L1 norm of the edited layer is identified as a key influence. Building upon these insights, the paper proposes a method named Dump for sequence (D4C), which significantly improves model performance.

**Strengths:**

- The paper is well-motivated: Exploring the reasons behind and impact of small changes made by model editing techniques on the performance of unedited samples is of great significance.
- The analysis of the data-specific and model-specific factors is supported with diverse datasets and comprehensive experiments. The model-specific analysis, in particular, is evaluated rigorously, addressing the forgetting issue that prior works often overlooked

- The observation of the influence of editing on the model norm is intriguing. High-norm parameters can be sensitive to noise and numerically unstable. It would be beneficial if the authors could also provide an L2-norm plot for comparison.

- The experimental results are impressive, demonstrating significant improvements and validating the effectiveness of the proposed method.

**Weaknesses:**

- My main concern with the data-specific analysis is whether the conclusion is about correlation or causation. Many variables can be changed about the input data. Plotting a single Figure 3 might be insufficient to justify that perplexity and token length are the main reasons for the decline in model performance after editing.

- Unfortunately, the constructed dataset is not open-sourced.

- Recent research [1] has shown that model editing methods (e.g. ROME, MEMIT) are not good at handling multi-hop questions, how would D4C perform in such more challenging scenarios?

- Some theoretical analysis can be conducted to demonstrate that D4C does not lead to an increase in norms.

[1] Mquake: Assessing knowledge editing in language models via multi-hop questions. EMNLP 2023

**Questions:**

- Can the authors add a section in the appendix to expand on the dataset mentioned in 3.1 (i.e., provide examples and details about the editing objectives) for better readability?

- What dataset was employed in Section 5?

- I encourage the authors to release the full code to enhance reproducibility.

- (Minor) Consider reducing v-space in some parts of the paper (e.g., the bottom of page 2).

**Limitations:**

The limitations are discussed in the paper.

---

> ### Author Rebuttal · Authors · 2024-08-05
>
> Thank you for recognizing the importance, effort in method, and applications of our work. We outline our response to the main concerns:
>
> $\textbf{Q1: }$ Can the authors add a section in the appendix to expand on the dataset mentioned in 3.1
>
> $\textbf{W1: }$ Thank you for your suggestion. We will provide a detailed introduction to the dataset in the appendix. For example, for the DG category in Table 1, the main structure of the dataset is as follows:
>
> |DG category |   |  |
> |----  |----  |----  |
> |Keys: | Text | Annotation  |
> | Prompt:| {}, resulting in PersonY, | Component Prompt
> |Subject:| PersonX accepts PersonY appointment | Edited Subject
> |Relation_id:| resulting in | Relation Category
> |Target_new:| {‘str’:shakes PersonX hand,} | Target Answer
>
> $\textbf{Q2: }$ The data-specific analysis is whether the conclusion is about correlation or causation.
>
> $\textbf{W2: }$  The conclusion is about causation. We echo your observation regarding the potential variability in input data variables. Our research found that perplexity and token length are the reasons for the performance degradation of the edited model. In order to control variables, as shown in Table 1, our 'Subject' and 'Relation_id' are consistent, that is, 'PersonX accept PersonY appointment, resulting in PersonY, '.  To make the model output answers that match the question type, the prompt will be slightly different. Each question type consists of 4000 samples.  Although it is difficult to exclude all irrelevant variables, it can be certain that the experimental results have demonstrated the validity of the conclusion, which has a positive effect on exploring the reasons for the decline in model performance after editing.
>
> $\textbf{Q3: }$ Recent research [1] has shown that model editing methods (e.g. ROME, MEMIT) are not good at handling multi-hop questions, how would D4C perform in such more challenging scenarios?
>
> $\textbf{W3: }$ Our work focuses more on sequence editing, which is a common and important scenario in real-life applications. Therefore, we did not explore multi-hop questions before.  Inspired by your suggestion, we conducted experiments on the multi-hop dataset Mquake [2]. The experimental results showed that our method also achieved superior performance.
>
>  |Llama-Method|Eff.|Par.|Mul.|Avg.|
> |----  |----  |----  |----  |----  |
> |FT|41.43|44.93|**28.24**|38.20|
> |ROME|`76.85`|**78.41**|3.67|`52.97`|
> |MEMIT|0.00|0.00|0.00|0.00|
> |D4C|**85.30**|`72.68`|`28.16`|**62.05**|
>
> |GPT-Method|Eff.|Par.|Mul.|Avg.|
> |----  |----  |----  |----  |----  |
> |FT|`17.00`|`6.22`|`0.00`|`7.74`|
> |ROME|0.00|0.00|`0.00`|0.00|
> |MEMIT|0.00|0.00|`0.00`|0.00|
> |D4C|**97.40**|**75.54**|**21.84**|**64.93**|
>
> PS: **Bold** indicates the best results, while suboptimal is `highlighted`. Since GRACE [1] is not suitable for the data structure of Mquake, we did not include it in the comparison.
>
> $\textbf{Q4: }$ What dataset was employed in Section 5?
>
> $\textbf{W4: }$ For our sequence edting experiment, we use the ZsRE [3] dataset. **And we also expanded our experiment. You can find the details in the Global Response.**
>
> $\textbf{Q5: }$ I encourage the authors to release the full code to enhance reproducibility.
>
> $\textbf{W5: }$  Thank you for your suggestion! We have included our code in the Supplementary Material when submitting the paper. You can download the file and follow the instructions in the README.md to set up the code.
>
> $\textbf{Q6: }$ Some theoretical analysis can be conducted to demonstrate that D4C does not lead to an increase in norms.
>
> $\textbf{W6: }$ For simplicity, we can consider the update of parameters edited by MEMIT after $n$ edits:
> $$\Delta W_{MEMIT}=\sum_{i=1}^{n} (r_ik_i^T)(K_0K_0^T+k_ik_i^T)^{-1}$$
> Regarding the D4C method, we have:
> $$\Delta W_{D4C}=(\sum_{i=1}^{n} r_ik_i^T)(K_0K_0^T+\sum_{i=1}^{n} k_ik_i^T)^{-1}=\sum_{i=1}^{n} (r_ik_i^T)(K_0K_0^T+\sum_{i=1}^{n} k_ik_i^T)^{-1}$$
>
> Due to $K_0K_0^T+k_ik_i^T$ and $K_0K_0^T+\sum_{i=1}^{n} k_ik_i^T$ being positive definite, intuitively, the inverse of $K_0K_0^T+\sum_{i=1}^{n} k_ik_i^T$ is expected to have smaller numerical values compared to $K_0K_0^T+k_ik_i^T$. Therefore, the norm of $\Delta W_{D4C}$ is smaller than that of $\Delta W_{MEMIT}$. We are considering adding a detailed theoretical analysis in the revised version.
>
> $\textbf{Q7: }$ It would be beneficial if the authors could also provide an L2-norm plot for comparison.
>
> $\textbf{W7: }$ Thank you for acknowledging our analysis. We have added comparative experiments on L2-norm, and the results are as follows
>
> |GPT2-ROME  |0|100|200|300|400|500|600|700|800|900|
> |----  |----  |----  |----  |----  | ----  |----  |----  |----  |----  | ---- |
> |L1-norm ( e-2 ) |1.06 | 1.58 | 2.20 | 2.71 |  3.14 | 3.52 | 3.87 | 4.19 | 4.49 | 4.77 |
> |L2-norm ( e-6 ) |1.67|  2.54|  3.68| 4.62| 5.40|  6.10|  6.71| 7.28|  7.82| 8.31|
>
> |GPT2-D4C  |0|100|200|300|400|500|600|700|800|900|
> |----  |----  |----  |----  |----  | ----  |----  |----  |----  |----  | ----  |
> |L1-norm ( e-2 ) | 1.06 | 1.09 | 1.00 | 1.11 |  1.11 | 1.11 | 1.11 | 1.11 | 1.12 | 1.12 |
> |L2-norm ( e-6 ) | 1.67|  1.68|  1.69| 1.70| 1.70|  1.71|  1.72| 1.73|  1.73| 1.74|
>
> The results showed that the conclusions of L2 norm and L1 norm are consistent. Our D4C method can effectively suppress norm growth.
>
> $\textbf{Ref}$
>
> [1] Aging with GRACE: Lifelong Model Editing with Discrete Key-Value Adaptors. Neurips 2023.
>
> [2] Mquake: Assessing knowledge editing in language models via multi-hop questions. EMNLP 2023.
>
> [3] Zero-shot relation extraction via reading comprehension. CoNLL 2017.

---

> ### Comment · Reviewer_MFeE · 2024-08-12
>
> I appreciate the authors' efforts in addressing my questions and am generally satisfied with the responses provided. However, I would like to request additional clarification regarding the connection with the paper [1], which already received 23 citations to date. This paper shares the same motivation as yours—addressing performance degradation—and also proposes regularization as a solution. While I see the references in the related work, the novelty appears somewhat overlapping. Could the authors further elaborate on how their work differentiates itself from this reference and justify the uniqueness of their approach in light of this existing work?
>
> [1] https://arxiv.org/abs/2401.04700v3

---

> ### Author Response · Authors · 2024-08-13
>
> Thank you for your prompt response. Below, we elaborate on the key distinctions between our work and the referenced studies [1], highlighting our novel contributions and advantages.
>
> * **Novelty:** Our research stands apart by pinpointing norm growth as the primary culprit behind model performance degradation, a revelation that precedes the publication of the V3 [1] version of the references. Notably, at the time of our NeurIPS 2024 submission deadline, only the V2 [2] version was available, which merely acknowledged a decline in model performance post-editing without delving into the underlying causes or proposing solutions. By contrast, we are the pioneers in identifying and addressing this critical issue.
>
> * **Evaluation Methodology:** Even if the V3 version is subsequently released, our work retains significant advantages. Specifically, we identified a fundamental flaw in V3's evaluation approach, which relies solely on the original single edit (e.g. evaluating the current editing performance after editing one sample). This approach fails to align with the standard experimental setup for sequence editing, where the performance of all previously edited samples is assessed after multiple edits. Our Figure 7 visually demonstrates the limitations of V3's method and underscores its inability to validate the resolution of catastrophic forgetting.
>
> * **Addressing Editing Bottlenecks:** Our paper goes beyond merely acknowledging the existence of editing bottlenecks, as exemplified by the 850-times limitation in Memit. We boldly extended our experiments to 10,000 edits, demonstrating the remarkable resilience of our model. In stark contrast, V3's experiments were confined to a mere 20 edits (as shown in Figure 7), indicating a lack of depth in exploring and addressing the editing bottleneck challenge. Furthermore, the disparity in evaluation methodologies and the limited scope of V3's edits hinder a fair performance comparison.
>
> * **Mitigating Norm Growth:** Both our work and V3 recognize norm growth as a pivotal factor in performance degradation. However, we present compelling evidence in Figure 6 that our D4C method effectively curbs norm growth, thereby safeguarding model performance. In contrast, V3's regularization strategy, which involves setting certain parameters to zero, lacks empirical support for its ability to significantly alleviate norm growth issues.
>
> Thank you again for your reply. If you have any remaining concerns or need further clarification, we welcome your additional input. Thank you for your continued consideration.
>
> **Ref**
>
> [1] V3 https://arxiv.org/pdf/2401.04700v3
>
> [2] V2 https://arxiv.org/pdf/2401.04700v2

---

> ### Comment · Reviewer_MFeE · 2024-08-13
>
> Thank you for providing such thorough and compelling responses. Despite the existing critiques on knowledge editing, this paper offers the most intuitive and straightforward demonstration of performance degradation, along with effective solutions. I am confident that this work will significantly advance the field, and I will be increasing my score to further support it.
>
> However, I would like to note that the current code provided in the supplementary materials is incomplete. For example, the MQD dataset that was constructed is not included. Ensuring the code's completeness and improving its visibility would likely enhance the impact of this work even further.

---

> > ### Author Response · Authors · 2024-08-13
> >
> > Thank you for your reply. We promise to publicly release the dataset and complete code in the future.

---

### Official Review · Reviewer_rpPG · 2024-07-12

**Soundness:** 3
**Presentation:** 3
**Contribution:** 3
**Rating:** 7
**Confidence:** 3

**Summary:**

The paper investigates the reasons behind performance decline in sequential model editing approaches that selectively update parameters based on both data and model factors. To address the issues causing this decline, the authors propose a method to save editing history, thereby transforming sequential editing into batch editing with minimal computational overhead.

**Strengths:**

Extensive experimentation is conducted to empirically demonstrate how factors such as dataset characteristics, editing objectives, and model-specific properties affect performance in sequential model editing.

A simple matrix storage solution is introduced, which enables the conversion of sequential editing into batch editing.

**Weaknesses:**

The study is restricted to two closely related editing approaches.

Experimentation is limited in demonstrating the efficacy of the D4C method. Different datasets and a larger number of edits for a more thorough evaluation are needed.

**Questions:**

N/A

---

> ### Author Rebuttal · Authors · 2024-08-05
>
> Thank you for your valuable feedback and for recognizing the novelty of the our method. Below, we address some of the weaknesses raised:
>
> $\textbf{Q1: }$ The study is restricted to two closely related editing approaches.
>
> $\textbf{W1: }$ First and foremost, our innovation surpasses the mere translation of sequence editing into batch editing. It employs a batch editing-like approach to implement sequence editing, resulting in a significant performance boost. As indicated in Table 2, previous methods like MEMIT [1] and PMET [2] were designed for batch editing, but they struggled to perform well when applied to sequence editing tasks. In contrast, as we highlighted in L173-L176, our focus on sequence editing aligns more closely with real-world requirements, highlighting the limitations of existing knowledge editing techniques. By enhancing the performance of sequence editing while maintaining a storage complexity of $O(1)$, a succinct theoretical proof is presented in Reviewer MFeF's 'Q6'. We aim to advance the practical application of knowledge editing technology.
>
> Furthermore, we conducted additional experiments by extending the baseline method with the latest state-of-the-art method, GRACE [3]. It is worth noting that although the GRACE method performs 22% worse than ours in performance, due to their storage complexity being $O(n)$, they did not provide experiments for editing 10,000 times.
>
> |GPT-J-ZsRE (1,000 times)|Eff.|Par.|Spe.|Avg.|
> |----  |----  |----  |----  |----  |
> |ROME|2.44|`2.23`|0.92|1.86|
> |MEMIT|0.00|0.02|0.00|0.01|
> |GRACE|**100.00**|0.07|**100.00**|`66.69`|
> |D4C|`95.98`|**91.17**|`70.17`|**88.77**|
>
> PS: **Bold** indicates the best results, while suboptimal is `highlighted`.
>
> $\textbf{Q2: }$ Experimentation is limited in demonstrating the efficacy of the D4C method. Different datasets and a larger number of edits for a more thorough evaluation are needed.
>
> $\textbf{W2: }$ Thanks again for your advice!  We further demonstrate the effectiveness of the method by adding additional datasets and number of edits. Firstly,  we expanded the datasets to include the Mquake [4] and Counterfact [1] datasets, where Mquake is a multi-hop dataset, Counterfact is a fact dataset. The results are as follows:
>
> |Llama2-Mquake (1,000 times)|Eff.|Par.|Mul.|Avg.|
> |----  |----  |----  |----  |----  |
> |FT|41.43|44.93|**28.24**|38.20|
> |ROME|`76.85`|**78.41**|3.67|`52.97`|
> |MEMIT|0.00|0.00|0.00|0.00|
> |D4C|**85.30**|`72.68`|`28.16`|**62.05**|
>
>
> |Llama2-Counterfact (1,000 times) |Eff.|Par.|Spe.|Avg.|
> |----  |----  |----  |----  |----  |
> |FT|8.46|4.07|2.03|4.85|
> |ROME|27.83|`16.03`|5.66|16.50|
> |MEMIT|0.00|0.00|6.72|2.24|
> |GRACE|**99.9**|0.25|**99.97**|`66.71`|
> |D4C|`96.68`|**46.66**|`72.45`|**71.93**|
>
> PS: **Bold** indicates the best results, while suboptimal is `highlighted`.
>
> Additionally, we extended the editing number of the our method to 10,000 (Llama2) and 9,000 (GPT). We evaluated the performance of edited GPT models in downstream tasks. As shown in the table below, the edited model maintains great performance on downstream tasks, proving that the editing method has minimal damage to the model.
>
> |Num of Edits|arc_challenge|hellaswag|mmlu|truthfulqa_mc|winogrande |
> |---- |----  |----  |----  |----  |----  |
> |0|43.34|57.14|41.35|38.97|69.14|
> |10,000|42.41 ($\downarrow 0.93$)|52.99 ($\downarrow 4.15$)|39.76 ($\downarrow 1.59$)|38.56 ($\downarrow 0.41$)|68.75 ($\downarrow 0.39$)|
>
> **Specific results can be seen in Fig. 1 and Fig. 2 of supplementary PDF and our Global Response.**  The experimental results show that our method can still achieve superior performance after 10,000 edits.
>
> $\textbf{Ref}$
>
> [1] Mass-editing memory in a transformer. ICLR 2023.
>
> [2] PMET: Precise Model Editing in a Transformer. AAAI 2024.
>
> [3] Aging with GRACE: Lifelong Model Editing with Discrete Key-Value Adaptors. Neurips 2023.
>
> [4] Mquake: Assessing knowledge editing in language models via multi-hop questions. EMNLP 2023.

---

> > ### Comment · Reviewer_rpPG · 2024-08-13
> >
> > Thank you for your detailed response, I have updated the score, the additional experiments should be added to the revised draft to show the efficacy of the approach.

---

> > > ### Author Response · Authors · 2024-08-14
> > >
> > > Thank you for your reply. We promise to add additional experiments in the revised version.

---

### Official Review · Reviewer_nwwK · 2024-07-13

**Soundness:** 2
**Presentation:** 2
**Contribution:** 2
**Rating:** 6
**Confidence:** 3

**Summary:**

This paper investigates the reasons and solutions for the decline in model performance of model editing.  The authors conduct experiments from two perspectives: data and model. Specifically, to clarify the impact of data on the performance of edited models, the authors first evaluate how editing different types of data affects model performance. Then, the authors construct a Multi-Question Dataset (MQD) and identified that the performance of  the edited models is primarily influenced by the diversity of the editing objectives  and the length of the tokens. Secondly, the authors explore the factors that affect model  performance from a model perspective. Experiments revealed a strong correlation between the L1 norm of the edited model layers and the editing accuracy, and  identified an editing quantity bottleneck. To enhance the performance of edited  models, the authors propose a Dump for sequence (D4C) method that effectively improves  the performance of edited models and overcomes the previous editing bottleneck issue. This method allows for multiple effective edits with minimal impact on  model performance.

**Strengths:**

This paper investigates the impact of data on the performance of edited models. Evaluations are conducted across multiple tasks, revealing that the editing objective is the primary factor influencing model performance.

The authors found that the decline in edited model performance is correlated with the explosive growth of the L1 norm of parameter layers during the editing process.

This paper proposes a caching sequence edit method that leverages O(1) space complexity to retain past knowledge and regulate the explosive growth of the parameter layer norm.

**Weaknesses:**

The writing of this paper should be improved. There is no overview of this paper, which makes it hard to follow the details of Section 3 and 4.

The motivation of the proposed method  is not clear.

There are many typos such as line 182.

There are many missing references such as:

Knowledge Editing for Large Language Models: A Survey

Stable Knowledge Editing in Large Language Models

A Comprehensive Study of Knowledge Editing for Large Language Models

Editing Large Language Models: Problems, Methods, and Opportunities

**Questions:**

See weaknesses.

**Limitations:**

Yes

The limitations are on page ten. I am unsure if this counts as exceeding the page limit.

---

> ### Author Rebuttal · Authors · 2024-08-05
>
> Thank you for the positive recommendations and valuable feedback!
>
> $\textbf{Q1: }$ There is no overview of this paper, which makes it hard to follow the details of Section 3 and 4.
>
> $\textbf{W1: }$  We provided an overview of this paper using text and figures. Firstly, in the Introduction section, L36-L43 and L44-L50 briefly introduce the relevant content of Sections 3 and 4, respectively. At the same time, we visually present the overview of this paper in Figure 1, including both data and model perspectives. Finally, in L93-L94 and L168-L169, we provide overviews of Sections 3 and 4, respectively.
>
> $\textbf{Q2: }$ The motivation of the proposed method is not clear.
>
> $\textbf{W2: }$ The motivation for this paper is to investigate the causes of performance degradation in edited models and to optimize them. We have explicitly expressed it multiple times in the paper. Firstly, in L4-L5 of the Abstract section, we present this motivation and provide our proposed solution. In addition, in L36-L37, we reiterated this motivation and provided specific experimental settings. Meanwhile, L51-L54 and L62 have repeatedly demonstrated this motivation. Finally, in the conclusion section, L280 and L283 express the motivation behind this paper.
>
> $\textbf{Q3: }$ About Limitations section.
>
> $\textbf{W3: }$ We learned from the NeurIPS website [1] that the Limitations are not included in the main text of the paper, so it can be placed on page ten.
>
> $\textbf{Q4: }$ About  some  missing references and typos.
>
> $\textbf{W4: }$ We promise to add missing references and correct typos in the revised version.
>
> $\textbf{Ref}$
>
> [1] CallForPapers. Neurips 2024.

---

> > ### Comment · Reviewer_nwwK · 2024-08-13
> >
> > Thank you for your reply, my concern has been addressed. I have raised my score.

---

### Author Rebuttal · Authors · 2024-08-06

# Global Response

We thank the reviewers for their thoughtful feedback. We are glad the reviewers find that

* Our motivation is innovation and has great significance

    * "The paper is well-motivated: Exploring the reasons behind and impact of small changes made by model editing techniques on the performance of unedited samples is of great significance." - MFeE
    * "This approach is both innovative and practical for managing edited models." -  Kq3N

* Our paper presents a thorough, novel, and significant analysis

    * "The model-specific analysis, in particular, is evaluated rigorously, addressing the forgetting issue that prior works often overlooked." -MFeE
    * "detailed analysis  provide valuable insights into the mechanics of model editing." - Kq3N

* Our solutions are novel and effective and our experiments are well-conducted

    * "The experimental results are impressive, demonstrating significant improvements and validating the effectiveness of the proposed method." - MFeE
    * "Extensive experimentation is conducted to empirically demonstrate." - rpPG
    * "The experiments enhance the credibility and applicability of the findings." -  Kq3N

**[Main Motivation]**

As the scale of models continues to grow, how to update model knowledge affordably has remained a persistent challenge. Despite recent intriguing proposals, such as knowledge editing [1], these methods have yet to see practical implementation. The primary issue lies in the significant drop in performance in the edited model, leading to a lack of trust from users. Understanding the causes behind this decline in model performance post-editing is increasingly crucial.
By deriving solutions from the analysis findings and minimizing model disruptions during knowledge updates, the edited model can earn user trust and drive the adoption of this technology in real-world scenarios. As highlighted by reviewer MFeE, this work holds considerable significance.


**[Supplementary Experiments]**

 *  **1. Dataset Expansion Experiment.**  We conducted additional experiments and expanded the datasets to encompass the Counterfact [1] and Mquake [3] datasets. The experimental outcomes for the Counterfact dataset are outlined below

    |Llama-Method|Eff.|Par.|Spe.|Avg.|
    |----  |----  |----  |----  |----  |
    |FT|8.46|4.07|2.03|4.85|
    |ROME|27.83|`16.03`|5.66|16.50|
    |MEMIT|0.00|0.00|6.72|2.24|
    |GRACE|**99.9**|0.25|**99.97**|`66.71`|
    |D4C|`96.68`|**46.66**|`72.45`|**71.93**|

    |GPT-Method|Eff.|Par.|Spe.|Avg.|
    |----  |----  |----  |----  |----  |
    |FT|32.80|9.00|1.00|14.27|
    |ROME|0.60|0.70|0.60|0.63|
    |MEMIT|86.20|**59.50**|31.30|59.00|
    |GRACE|**100.00**|0.50|**100.00**|`66.83`|
    |D4C|`99.10`|`47.00`|`63.70`|**69.93**|

    PS: **Bold** indicates the best results, while suboptimal is `highlighted`.

    Additionally, taking into account Reviewer MFeE's suggestions, we investigated the performance of existing baselines under **sequence multi-hop editing** with Mquake [3] dataset:

    |Llama-Method|Eff.|Par.|Mul.|Avg.|
    |----  |----  |----  |----  |----  |
    |FT|41.43|44.93|**28.24**|38.20|
    |ROME|`76.85`|**78.41**|3.67|`52.97`|
    |MEMIT|0.00|0.00|0.00|0.00|
    |D4C|**85.30**|`72.68`|`28.16`|**62.05**|

    |GPT-Method|Eff.|Par.|Mul.|Avg.|
    |----  |----  |----  |----  |----  |
    |FT|`17.00`|`6.22`|`0.00`|`7.74`|
    |ROME|0.00|0.00|`0.00`|0.00|
    |MEMIT|0.00|0.00|`0.00`|0.00|
    |D4C|**97.40**|**75.54**|**21.84**|**64.93**|

    PS: Since GRACE is not suitable for the data structure of Mquake, we did not include it in the comparison.

* **2. Method expansion experiment.**  We conducted further experiments by extending the baseline method with the latest state-of-the-art method, GRACE [2]. Notably, while the GRACE method shows a performance decrease of only 22% compared to ours, their storage complexity being $O(n)$ prevented them from conducting experiments involving 10,000 edits.

    |GPT-1000 times|Eff.|Par.|Spe.|Avg.|
    |----  |----  |----  |----  |----  |
    |ROME|2.44|`2.23`|0.92|1.86|
    |MEMIT|0.00|0.02|0.00|0.01|
    |GRACE|**100.00**|0.07|**100.00**|`66.69`|
    |D4C|`95.98`|**91.17**|`70.17`|**88.77**|


* **3. Editing frequency expansion experiment.** we extended the number of edits to 10,000, and the edited model still maintained great performance.

    |Num of Edits|arc_challenge|hellaswag|mmlu|truthfulqa_mc|winogrande |
    |---- |----  |----  |----  |----  |----  |
    |0|43.34|57.14|41.35|38.97|69.14|
    |10,000|42.41 ($\downarrow 0.93$)|52.99 ($\downarrow 4.15$)|39.76 ($\downarrow 1.59$)|38.56 ($\downarrow 0.41$)|68.75 ($\downarrow 0.39$)|

    The existing knowledge editing methods have poor editing performance (such as Table 2), or have a storage complexity of $O(n)$ (e.g. GRACE [2]), and cannot provide experimental results edited 10,000 times.

**The supplementary PDF provides more detailed experimental results.**

$\textbf{Ref}$

[1] Knowledge editing for large language models: A survey.  arXiv 2023.

[2] Aging with GRACE: Lifelong Model Editing with Discrete Key-Value Adaptors. NeurIPS 2023

[3] Mquake: Assessing knowledge editing in language models via multi-hop questions. EMNLP 2023

---

### Author Response · Authors · 2024-08-12

Dear Senior Program Committee and Reviews,

We appreciate your initial feedback and have addressed your comments in our previous response. If you have any remaining concerns or need further clarification, we welcome your additional input. Thank you for your continued consideration.

Sincerely Yours

The Authors

---

### Decision · Program_Chairs · 2024-09-25

**Decision:**

Accept (poster)

**Comment:**

The paper analyzes the reasons of decline in model's performance after editing. They discuss both data and model aspects and provide various insights. Moreover, they suggested a method to enhance model's performance. The paper contributes to a large space of model editing methods and is a valuable contribution.